# The Relationship between Training Load and Injury Risk in Basketball: A Systematic Review

**DOI:** 10.3390/healthcare12181829

**Published:** 2024-09-13

**Authors:** Chi-Chung Chan, Patrick Shu-Hang Yung, Kam-Ming Mok

**Affiliations:** 1Department of Orthopaedics and Traumatology, The Chinese University of Hong Kong, Hong Kong, China; 1155155742@link.cuhk.edu.hk (C.-C.C.); patrickyung@cuhk.edu.hk (P.S.-H.Y.); 2School of Interdisciplinary Studies, Lingnan University, Hong Kong, China

**Keywords:** injury prevention, strength and conditioning, sports medicine, sports coaching, sports injury

## Abstract

The relationship between training load and injury risk in basketball is an important area in sports injury prevention and performance enhancement; however, there is limited conclusive evidence of their associations. The aim of this systematic review was to examine the evidence of the relationship between training load and injury risk in basketball, which is one of the most common sports worldwide. This systematic review was conducted according to the Preferred Reporting Items for Systematic Reviews and Meta-Analyses guidelines. A comprehensive literature search was conducted on the PubMed, SCOPUS, and Web of Science databases up until March 2024. The search aimed to identify studies that prospectively and/or retrospectively examined the relationship between training load and injury risk in basketball. Inclusion criteria were limited to studies published before February 2024. The quality of each study was assessed using the Newcastle–Ottawa Quality Assessment Scale and Oxford Centre for Evidence-Based Medicine levels of evidence. A narrative synthesis of the findings was performed. A total of 14 articles met the inclusion criteria and were included in the review. Of these, 11 studies reported at least partially statistically significant results, providing evidence of a relationship between training load and injury risk. In conclusion, the findings of this review suggest a clear relationship between training load and injury risk in basketball.

## 1. Introduction

Load management, including prescription, monitoring, and adjustment of external and internal loads, is essential for enhancing performance and reducing injury risk for athletes [1]. In basketball, there was around 10 times higher injury incidence during competition than during training in professional basketball [2]. There have been several changes in load management policy in three major basketball leagues around the world in recent years, including the National Basketball Association (NBA), International Basketball Federation (FIBA), and National Collegiate Athletic Association (NCAA). The International Basketball Federation introduced the Players First team competition system for men to restrict the game times; as a result, players have had a 26% reduced workload during the Olympic cycle since 2017. In addition, the NCAA have monitored the training and days off during and outside the in-season under time restrictions and management in legislation involving health and safety issues since June 2017. They were trying to optimize the training prescription competition load to avoid overloading [3].

In the NBA, greater performance load and fatigue was associated with increased risk of injury [4]. Moreover, there was a potential relationship between training load and injury incidence as injury incidence increased throughout the season [5]. Similarly, in a later study conducted by Torres-Ronda et al. [6], most injuries and missed games because of injury happened from mid-season to the end of the regular season [6]. Calendar congestion was another issue affecting performance and injury [7]. Playing back-to-back games had significantly greater number of games lost compared to having at least one day of rest [7]. Meanwhile, the accumulation of fatigue and incomplete physical recovery caused by the calendar congestion could have a season-long impact on the home advantage [8]. On top of the detrimental effect on team performance, a significantly higher incidence of injuries was found in the condensed 2020 season compared to the previous four seasons after the adjustment of the COVID-19-related absences, consistent with the situation in other congested seasons of 1998 and 2011 [9]. However, there was no association between playing four games in five days alone and the increased rate of game injuries [10].

The International Olympic Committee published a consensus statement on load in sport and risk of injury in 2016. The statement provided guidelines for prescription of training and competition load as well as monitoring of training, competition, psychological load, athlete well-being, and injury [1]. Several reviews were published based on a limited number of sports such as rugby, soccer, and Australian football only in the male population [11,12,13]. In 2018, according to the systematic review related to load injury in sports, Eckard and his colleagues [11] concluded the existence of a relationship between training load and injury was continuously well-supported, confirming the previous reviews from Jones et al. [12] and Drew and Finch [13]. In addition, there was an increasing number of systematic reviews taking account into inter- and intra-individual variation to be more sports-specific and injury-specific such as bowling workload and lower-back injuries in fast bowlers in cricket [14] as well as pitch velocity and ulnar collateral ligament in baseball [15]. Unfortunately, there was no systematic review examining the relationship between training load and injury risk in basketball.

The increasing number of original studies investigating the relationship between training load and injury risk in basketball reflected a growing concern. Despite the availability of information on training load and injury risk from other sports, there is insufficient knowledge about its influence on basketball. The primary objective of this systematic review was to provide an overview of the relationship between training load and injury risk in basketball. Moreover, the secondary objective was to identify the critical risk factor on load and injury. The hypothesis was that the relationship between training load and injury risk in basketball existed.

## 2. Materials and Methods

### 2.1. Literature Search

The present systematic review was conducted in accordance with the Preferred Reporting Items for Systematic Reviews and Meta-Analyses (PRISMA) guidelines. The review protocol has been registered in INPLASY (Registration number: INPLASY202470005). The search strategy involved a comprehensive search of the PubMed, SCOPUS, and Web of Science databases up until March 2024, using relevant keywords related to training load, musculoskeletal injury, and basketball, as presented in Table 1. To manage the retrieved articles, EndNote 20.6, a reference management software program, was utilized to organize the saved articles, and the total number of saved articles from each database was recorded both before and after the removal of duplicates.

### 2.2. Selection Criteria

The present study employed specific inclusion criteria to identify studies for analysis. Only prospective or retrospective cohort designs were considered, with the exclusion of case studies, case series, case–control studies, review papers, or purely epidemiology [11,13]. The study population consisted of basketball athletes participating at all competition levels. To be included, studies must have calculated at least one measure of training load, measured as external or internal, subjective or objective, absolute or relative, taken during training, competition, or strength and conditioning sessions. Training load was defined as the cumulative amount of stress placed on an individual from single or multiple training sessions (structured or unstructured) over a period of time [1]. Additionally, studies must have reported musculoskeletal injuries as an outcome, either self-reported or diagnosed by a physician, with the exclusion of illness and performance-related measures. Injury was defined as any physical complaint resulting from competition or training, regardless of its impact on sports participation or performance [1]. There were no geographic limitations, and only English studies published before March 2024 were included in the analysis. Incomplete studies, such as abstracts, conference proceedings, commentaries, editorials, and letters, were excluded. Studies that did not meet any one or more of these specific inclusion criteria were also excluded from the present investigation.

### 2.3. Quality Assessment

To assess the quality of each study, including the possibility of bias, the Newcastle–Ottawa Quality Assessment Scale (NOS) was utilized for cohort studies [16]. All three authors were involved in the assessment, the first author was responsible for the primary data extraction, the last author verified the data. If there was a disagreement, the second author would be involved in the discussion. This tool evaluates studies based on three areas, (1) participant selection, (2) comparability, and (3) outcome, and is specifically designed for non-randomized study designs. The Oxford Centre for Evidence-Based Medicine Model was used to determine the level of evidence for each study that met the inclusion criteria. The overall level of evidence was classified into five different levels: (1) strong, which indicates consistent findings among multiple high-quality randomized controlled trials (RCTs); (2) moderate, indicating consistent findings among multiple low-quality RCTs and/or non-randomized controlled trials (CCTs) and/or one high-quality RCT; (3) limited, which refers to one low-quality RCT and/or CCTs with conflicting evidence; (4) conflicting, indicating inconsistent findings among multiple trials (RCTs and/or CCTs); and (5) no evidence, which means that no RCTs or CCTs were available.

### 2.4. Data Extraction and Analysis

The investigator collected data from each article, including the year of publication, study design, location, population, sample size, and demographics. They also recorded the type of training or competition load data collected and the units in which it was reported. The study results were reported, including statistics such as odds ratios, correlations, mean differences, and injury rates. All data were collected using a standardized form, and a narrative synthesis was completed to report the findings on the relationship between training load and musculoskeletal injury. The number of articles using each type of training load was reported, along with comparisons within and between groups of studies using the same type of load. The strength of evidence was evaluated based on the number of studies examining each type of load, the proportion with significant findings, and the methodological quality of the studies.

## 3. Results

### 3.1. Article Identification

A total of 9301 potentials articles were identified through the initial search of the databases. After removal of 3622 duplicates as well as 236 books/book sections and conference proceedings, 5443 articles remained. The titles and abstracts of the articles were then reviewed, and 29 articles remained. Based on the inclusion and exclusion criteria, the full-text versions of these 29 articles were reviewed, resulting 14 articles included in the final dataset. A flowchart diagram summarizing the article identification process is shown in Figure 1.

### 3.2. Description of the Included Articles

All 14 of the included articles were cohort studies. The greatest number of articles originated from Europe (*n* = 6, 43%) [17,18,19,20,21,22] and North America (*n* = 6, 43%) [23,24,25,26,27,28], followed by Australia (*n* = 2, 14%) [29,30]. Most studies (*n* = 8) (57%) [17,18,20,22,25,26,27,29] had all male participants, three (21%) [21,24,28] had all females, and two (14%) [23,30] had both. One (8%) [19] article did not specific the gender of participants. The mean age range was 16.0–27.6 years with three [25,27,28] studies not reporting age. Most athletes from the studies (*n* = 9, 64.3%) [17,18,19,20,21,22,26,27,29] played the top level of basketball league in their regions, with three studies (21.4%) [24,25,28] including college level athletes and two studies (14.3%) [23,30] included high school level athletes accounting for the rest. Sample size ranged from 8 to 196, with seven studies (50%) [18,21,22,24,25,28,29] having less than 30 participants, six (42.9%) [17,19,20,23,26,30] having 30–50 participants, one (7.1%) [27] having more than 100 participants. Eight studies (57.1%) [18,21,22,23,24,28,29,30] monitored participants across 1 season (range from 16–34 weeks), two studies (14.3%) [17,25] across two seasons, two studies (14.3%) [20,26] across three seasons, and two studies (14.3%) across five seasons [27] and seven seasons [19], respectively. In terms of type of record, nine studies (64.3%) [17,18,19,21,22,23,25,28,29] monitored training and competition, three studies (21.4%) [20,26,27] monitored competitions, and two studies (14.3%) monitored training [24] and monitored physical activity [30], respectively. Article characteristics are presented in Table 2.

### 3.3. Definitions of Injury

The definition of injury varied across the included studies, with three studies (18.75%) [17,18,23] referring to the consensus statement of Fuller et al. [31] that defined injury as any physical complaint sustained by a player that results from a football match or football training, irrespective of the need for medical attention or time-loss from football activities. However, Ferioli et al. [17] only included non-contact injuries in their studies. Three studies (18.75%) [19,20,21] referred to the methodology of the Union of European Football Association (UEFA) consensus statement of Hagglund et al. [32], who defined an injury that occurred during a scheduled training session or match that caused absence from the next training session or match. In two studies (12.5%) [22,29] based on the Oslo Sports Trauma Research Center Questionnaire [33], an athlete reported reduced sports participation, training modifications, performance reductions, as well as symptoms. Two studies (12.5%) [24,25] reported time-loss injury by the team athlete trainer. Two studies (12.5%) [26,27] reported injury data from online databases Spotrac.com and Pro Sports Transactions, respectively. One study (6.25%) [28] reported that injury data were extracted from medical injury reports generated as injuries occurred. One study (6.25%) [30] defined an injury as an incident related to physical activity that resulted in either time lost from athletic participation, medical diagnosis and treatment from the study of Noyes et al. [34], or the presence of pain or discomfort.

### 3.4. Measures of Load

Training load was measured using internal loads (*n* = 3, 21.4%) [24,25,29], external loads (*n* = 5, 35.7%) [19,20,26,27,30], or both (*n* = 6, 42.9%) [17,18,21,22,23,28]. All studies except one study [29] utilized an absolute measure of training load. Two studies [17,29] used ACWR as a relative measure. One study [23] utilized both absolute and relative measure of training load. A total of eleven studies [17,18,19,20,21,22,23,26,27,28,30] measured at least one form of external load, with six studies [17,18,19,21,22,30] utilizing training and/or competition time, five studies [19,20,26,27,28] utilizing performance measures, three studies [18,19,21] utilizing training and/or competition frequency, two studies [26,27] utilizing minutes played per game (MPG), two studies [28,30] utilizing type of training/competition, two studies [22,23] utilizing neuromuscular function, and one study [20] utilizing speed and/or acceleration. In terms of internal load monitoring, eight studies [17,18,21,22,23,24,25,29] utilized sRPE. All these eight studies utilized sRPE multiplied by session duration, with seven studies [17,18,21,23,24,25,29] using the Borg CR-10 scale and one study [22] using the Borg CR-6–20 scale. Five studies [17,21,23,25,29] used the Borg CR-10-point scale adapted by Foster et al. [35], one study [17] used the Borg CR-10-point scale adapted by Borg [36], one study [24] used a modified Borg 1–10 scale adapted by Foster [37], and one study [22] utilized a modified Borg 6–20 scale adapted by Foster et al. [35]. Studies also utilized RPE (*n* = 3) [17,18,21], recovery and stress questionnaires/ratings (*n* = 3) [22,25,28], sleep (*n* = 2) [22,28], as well as ACWR (*n* = 2) [17,29]. Overall, eleven studies [18,19,20,21,23,24,25,26,27,28,29] had at least partially statistically significant results, demonstrating a relationship between training load and injury risk, with seven studies [18,19,20,23,24,26,27] reporting a direct relationship between load and injury risk, two studies [21,25] reporting both inverse and direct relationships, and two studies [28,29] reporting a U-shaped relationship. However, three studies [17,22,30] reported no significant relationship between load and injury risk. Training load in basketball was measured using internal loads, external loads, or both, with the majority of studies utilizing absolute measures and some incorporating the ACWR as a relative measure. Overall, most studies found a relationship between training load and injury risk, with some indicating a direct relationship and others showing both direct and inverse relationships, while a few studies reported no significant relationship.

### 3.5. Assessment of Article Quality, Level of Evidence, and Conflict of Interest

The median score for the overall Newcastle–Ottawa Scale (NOS) was 6, with a range of 4 to 8. The median score for participant selection was 3, with a range of 2 to 4. The median comparability score was 1, with a range of 0 to 2, while the median outcome score was 2, with a range of 1 to 3. The scores on the NOS can be categorized into three levels of article quality, namely, “good”, “fair”, and “poor”, according to the guidelines of the US Agency for Healthcare Research and Quality (AHRQ). Ten articles were rated as “good”, and four articles were rated as “poor”. According to the Oxford Centre for Evidence-based Medicine Model, nine articles were considered level 2b evidence and five articles were considered level 4 evidence. Scores for each article were presented in Table 3. Two of the included studies [23,29] reported receiving funding to complete the research, three studies [19,20,26] reported no funding, and nine studies did not include a statement about funding. Seven studies reported no conflict of interest, six studies [22,24,25,28,29,30] did not include a statement about conflict of interest, and one study [26] reported a potential conflict of interest.

## 4. Discussion

The primary purpose of this systematic review was to examine the evidence for a relationship between training load and injury risk in basketball. Our findings largely agree with the consensus statement from International Olympic Committee as well as the previous systematic review on load and injury risk [1,11,12,13]. The results of this systematic review highlight the relationship between training load and injury risk in basketball as follows, although there were some limitations in generalizability of the results.

### 4.1. Load Monitoring in Basketball

The most widely used internal load measures in this systematic review was sRPE, which was consistent to the explorative systematic review of practices as well as the narrative review in basketball [38,39]. In elite basketball training session, the most appropriate training load parameters were acceleration and change in direction for centers, deceleration and high-intensity jumps (over 0.4 m) for guards, and high-intensity and total amount of deceleration (>3.5 m·s^−2^) and change in direction for forwards [40]. There was a high correlation between the total amount of acceleration, deceleration, as well as change of direction and RPE and sRPE for all positions in elite basketball training [40]. The sRPE was significantly correlated with the number of high-acceleration movements at >4G, >6G, and >8G in simulated matches; however, the magnitude of the correlation decreased as the acceleration threshold increased [41]. In elite basketball competition, all positions of players experienced higher number of maximal decelerations (>3 m/s^2^) than accelerations while the number of moderate accelerations (<3 m/s^2^) were higher than that of moderate decelerations [42]. Hence, according to the systematic review of training load and match-play demands in basketball based on competition level, elite-level players covered less distance at lower average velocities and with lower maximal and average heart rates during the games when compared to sub-elite and youth players [43]. The relationship between tracking variables and injury risk in professional male basketball games was assessed in one of the included studies [20]. Although the studies did not analyze the positional differences, the studies reported that players with less or equal than three decelerations (2 m/s^2^) per game had a higher risk of injury during games [20]. Therefore, the number of high-intensity decelerations played an important role in injury prevention under game monitoring. Moreover, professional male players who covered less or equal to 1.3 miles per game had a higher risk of injury [20]. Power forward had the lowest total external training loads during games [42]. Therefore, the power forward may be at a higher risk of injury due to a lower external workload during game, compared to other playing positions.

### 4.2. Training and/or Competition Time and Injury Risk

Five studies investigated the relationship between training and/or competition time and injury risk. The studies reported contrasting findings, with a negative relationship in two studies [19,21], positive relationship in one study [18], and no relationship in two studies [23,28]. Piedra et al. [21] found that there was a significant negative relationship between total training time and the number of time-loss injuries, as well as a possible association between exposure time and a lower risk of time-loss injury; hence, increasing specific exposure time could be associated with a decrease in the risk of time-loss injuries. Similar to the study conducted by Garcia et al. [18], there was a phenomenon that athletes with less accumulated minutes in training and competition suffered more microtrauma injuries, supported with low absolute load associating with an increased risk of injury [1]. However, Caparrós et al. [19] found that there was a positive correlation between exposure (total number of practices and hours of exposure) and the total number of injuries in professional male basketball; hence, increasing practice and competition time increased the number of injuries. On the other hand, Gianoudis et al. [30] found that there was no significant difference in the total amount of physical activity undertaken weekly by injured and uninjured athletes and Ferioli et al. [17] found that there was no association between total exposure and non-contact injuries among professional and semi-professional athletes, showing no injury predictive ability. The contrast findings may be explained by the difference between total training and competition time among the studies. The individual average season training and competition time were as follows: 102 h in Garcia et al. [18], 289 h in Piedra et al. [21], 359 h in Caparrós et al. [19], and 404 (D1)/439 (D2)/280 (D3) hours in Ferioli et al. [17] except school-level players in Gianoudis et al. [30]. Therefore, low training and/or competition time might increase the risk of injury in professional basketball.

### 4.3. Relative Load, Rapid Changes in Load, and Injury Risk

Two studies [17,29] used ACWR as a relative measure. The studies reported contrast findings with one [32] presenting the finding that maintaining ACWR of 1–1.5 may be optimal for reducing injury risk in professional players. The second study [17] stated that there was no association between ACWR and non-contact injuries in professional and semi-professional players and no injury predictive ability (AUC range: 0.494; Youden index range: 0.056) was shown. This may be explained by the difference of definition of injury. Weiss et al. [29] used a self-reported injury questionnaire while Ferioli et al. [17] used the consensus statement of Fuller et al. [31], which was the most common used definition for examining ACWR and injury risk in team sports [17,29,31,44], and only accounted for non-contact injuries. Moreover, one study [23] found that youth players with overuse injuries and any lower extremity injury had a low previous 3–4-week jump count, jump height, and weighted jump height coupled with a high previous one-week workload before getting injured, without calculating the ACWR. Same as the systematic review conducted by Andrade et al. [45], the risk of time-loss injury in professional team sports may be increased by a lower chronic load coupled with a high ACWR. In addition, Orringer and Pandya [26] found that increased MPG over the cumulative three, five, and ten games directly preceding injury were closely related to increased injury occurrence, supported with rapid increased in workload increasing the injury risk [1]. Therefore, ACWR seemed to be one of the key risk factors for injury in basketball.

### 4.4. Minutes Played Per Game (MPG) and Injury Risk

Two studies [26,27] investigated the relationship between MPG and injury risk among NBA players. Orringer and Pandya [26] found that increased MPG over the cumulative 3 games (26.26, 4.9% increase, *p* = 0.04), 5 games (26.52, 5.8% increase, *p* = 0.004), and 10 games (26.03, 4.0% increase, *p* = 0.02) were statistically significantly correlated with injury occurrence, compared to the season average (25.01). NBA players with 20.0–29.9 MPG (IPR, 1.56 [95% CI, 1.12–2.17]) and ≥30.0 MPG (IPR, 1.67 [95% CI, 1.47–1.90]) demonstrated significantly greater incidences of injury during condensed NBA season under the coronavirus 2019 (COVID-19) and NBA players played in 26–35 min had the higher percentage of unique injuries, irrespective of the season being before or during COVID-19 [6,9]. The MPG was significantly associated with season-ending injuries (odds ratio, 1.06, 95% confidence interval, 0.99–1.01, *p* < 0.001) [27]. MPG was also found to be a significant associated risk factor of knee [46] and ankle [47] injuries in NBA. However, the MPG in a single NBA game did not contribute to anterior cruciate ligament injury because a large number of injuries happened in the first quarter of the season, which may have been caused by insufficient conditioning [48]. Therefore, MPG seemed to be the critical risk factors of lower-limb injury in professional basketball during the season.

### 4.5. Sleep and Injury Risk

Two studies [24,27] investigated the relationship of sleep and injury risk among male and female collegiate players in the NCAA’s Division 1. The first study discovered that increased a 1 h of sleep duration was independently associated with a 43% decreased risk of in-season injury after the adjustment of training load and subjective well-being [25]. The second study discovered that poor sleep patterns such as rapid eye movement (REM) and respiratory rate (RR) were more prone to injuries [27]. REM sleep was the most significant contributors to injury (0.11 CORR, 2.7% XGB, 12.9% RFC), and low (<20%) and high (>30%) REM% increased the chances of injury [28]. The partial dependence plots (PDPs) also showed that outside the typical range of 12–18 repetitions per minute of RR, sleep disturbances increased, which increased the chances of injury [28]. Therefore, sleep duration and sleep pattern were associated with injury risk in collegiate basketball.

### 4.6. Competition Calendar Congestion and Injury Risk

Calendar congestion was associated with an increased injury risk [1], confirmed by the systematic review conducted by Page et al. [49], which showed an increased match injury incidence under congested fixture schedules (a minimum of 2 matches with ≤4 days recovery) in professional male soccer. One study [22] examined the relationship between load and injury under a congested game schedule in elite basketball. The result showed that there were no significant differences in severity scores in the self-reported injury questionnaire as well as time loss between short-term match congestion (≥2-match weeks) and regular competition (1-match week) [22]. However, the incidence of injuries was significantly higher in the condensed 2020 NBA season than in the previous four seasons, which was consistent with the other congested NBA seasons in 1998 and 2011 [9]. The conflicting results may be explained by overcompensation of training load of coach, leading by lower total and training load as well as better well-being and less fatigue of players during short-term match congestion [22]. In addition, game schedules may affect the risk of injury in professional basketball. There were no associations between playing back-to-back games or playing four games in five days and an increased number of injuries in the NBA; however, playing back-to-back games and away games resulted in more game injury [10,50]. Therefore, competition calendar congestion as well as game schedules may affect the injury risk in professional basketball.

### 4.7. Limitation

Although this systematic review had several strengths, there were some limitations. First, this review only included English articles and some Spanish papers were missed. Second, there may be some publication bias because case studies, case series, case–control studies, review papers, abstracts, conference proceedings, commentaries, editorials, and letters were excluded. Studies related to pure epidemiology were also excluded. In addition, the definition of injuries was different, and the types of training load were diverse among the studies. Moreover, most injuries were self-reported and non-contact types. Therefore, the comparison between the studies was hard. Lastly, the subject of the studies came from various levels and gender, which limited the generalizability of the results.

### 4.8. Practical Applications and Future Direction

For the future direction of the studies, more studies should be performed to find out the optimal training/competition loads or upper limits for different level and different age of players. Also, more studies should be conducted to confirm whether the sweet spot in ACWR existed in basketball and therefore to provide the optimal progressive increase in MPG for the players. Practically, evidence shows that increasing sleep duration and maintaining optimal sleep patterns, along with managing MPG, are key for injury prevention in basketball. Practitioners should regularly monitor training load and sleep, aim for an ACWR of 0.8–1.3, ensure athletes get 8 h of sleep, and progressively increase MPG based on the season average to reduce injury risk. In addition, practitioners or international basketball governing bodies should investigate the relationship between competition calendar congestion and travel load on injury in basketball. On the other hand, 3 × 3 basketball has become a more popular modern sport in the world; however, there was only one study found investigating the training load parameters in 3 × 3 basketball tournaments during the comprehensive search [49].

## 5. Conclusions

To conclude, there was clear evidence in relationship between training load and injury in basketball, especially in sleep for the internal load monitoring and MPG for the external load monitoring. There was evidence showing low chronic load increases injury and high chronic load may increase injury and rapid changes in load also increase injury. However, there was no evidence showing that maintaining ACWR of 1–1.5 reduce injury and condensed schedule may increase injury. The application of ACWR to real practices should be conservative. For practitioners, they should regularly monitor the training load and sleep for athletes. Maintaining ACWR of 0.8–1.3 and 8 h of sleep seems to be optimal based on the current evidence. In addition, they should manage the MPG of individual athlete to reduce the injury risk. Progressive increase MPG of athletes based on their season average seems to be optimal for injury prevention.

## Figures and Tables

**Figure 1 healthcare-12-01829-f001:**
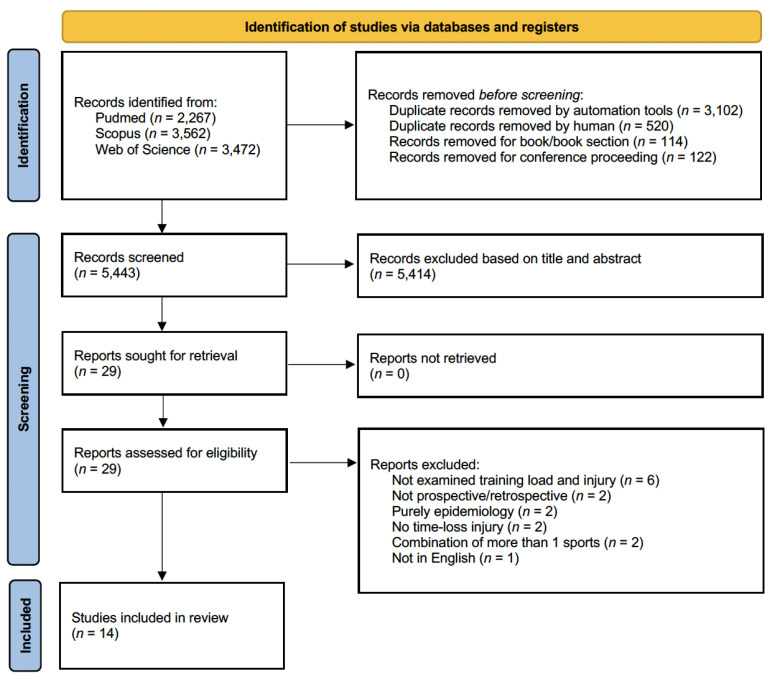
The flow of information through the systematic literature search and screening.

**Table 1 healthcare-12-01829-t001:** Variable and relevant keywords.

Variable	Search Strings
Training load and/or competition load	“load *” OR “workload *” OR “train *” OR “compet *” OR “recovery” OR “volume *” OR “intensit *” OR “duration *” OR “stress *” OR “congestion” OR “saturation” OR “distance” OR “exposure *” OR “hours” OR “days” OR “weeks” OR “jump *” OR “psychosocial *” OR “travel” OR “acute:chronic load ratio” OR “acute: chronic workload ratio” OR “ACWR” OR “exponentially weighted moving average” OR “EWMA” OR “perception of effort” OR “rating of perceived exertion” OR “RPE”

**Table 2 healthcare-12-01829-t002:** Summary of findings for studies investigating the relationship between training load and injury.

Articles	Participants	Injury Definition	Internal Load	External Load	Summary of Findings
Anderson et al. [24] Prospective USA 1 season	*n* = 12 NCAA D3 All female Age 18–22	An injury was defined as a circumstance in which the athlete received an evaluation from the team’s student athletic trainer and ATC and required limiting their practice for at least 1 day	sRPE Monotony Strain	Nil	A moderately positive correlation was found between weekly injuries and total weekly training load (*p* ≤ 0.01; r = 0.675) and between strain and monotony (*p* ≤ 0.01; r = 0.668) in A Pearson Product Moment correlation
Gianoudis et al. [30] Prospective Australia 1 season	*n* = 46 28 males 18 females High school Mean age 16.0	An injury was defined as an incident related to physical activity, that resulted in either time lost from athletic participation, medical diagnosis and treatment, or the presence of pain or discomfort	Nil	Duration of physical activity (hours per week)	No significant differences were found in the average weekly participation hours of physical activity of injured and uninjured players in independent t test (*p* = 0.67)
Caparrós et al. [19] Retrospective Spain 7 seasons	*n* = 44 F.C. Barcelona No gender information Age 27.6 ± 4.1	Time-loss injury: any injury occurring during a practice season or matches that caused an absence for at least the next practice session or match	Nil	Exposure time (no. and hours of games and practice)	A strong positive correlation between exposure (total number of practices and hours of exposure) and the total number of injuries in Pearson’s correlation (r = 0.77; *p* = 0.04)
Weiss et al. [29] Prospective Australia 1 season	*n* = 13 Australia New Zealand Basketball League All male Age 24.4 ± 4.7	Self-reported injury: Oslo Sports Trauma Research Center Injury Questionnaire	ACWR by sRPE (rolling average)	Nil	Proportions of injured squad members at workload ratios between 1.0–1.49 were substantially less than those observed at all other ratios by clear small to moderate amounts. Workload ratios ≤ 0.5, between 0.5–0.99, and ≥1.5 resulted in 1.5, 1.4, and 1.7 times more injured players, respectively. Comparisons between all other workload ratio ranges were trivial-to-small in magnitude and unclear, using the 90% CI to determine the significance
Caparrós et al. [20] Retrospective Spain 3 seasons	*n* = 33 Professional team All male Age 24.9 ± 2.9	Time-loss injury: any injury (contact and non-contact) occurring during a practice session or game that caused an absence for at least the next practice session or competition	Nil	Physiological variables Speed and distance variables Mechanical load variables Locomotor variables	A significant higher risk of injury during games were found in athletes with ≤3 decelerations with 2 m/s^2^ (IRR, 4.36; 95% CI, 1.78–10.6) and those running ≤ 1.3 miles (lower workload) (IRR, 6.42; 95% CI, 2.52–16.3) (*p* < 0.01 in both cases)
Piedra et al. [21] Prospective Spain 1 season	*n* = 11 Women’s league 1 All female Age 23.36 ± 2.99	Muscular pain/injuries required attention of the team physiotherapist/time-loss injury: any injury that occurred during training or a game and that led to the absence for at least the following session or game	RPE sRPE	No. of training practices No. of games Total hours of exposure Hours of exposure to games Hours of exposure to training practices	Several significant differences were observed between the injury risk values and the morning RPE (F = 5.0811; *p* = 0.032), the sRPE of the morning practices (F = 7.3585; *p* = 0.010) and the total time of exposure (F = 3.5055; *p* = 0.064) in the one-way ANOVA test. Significant negative relationship was observed between total training time and the number of time-loss injuries (rho = −0.797; *p* = 0.003) in the Spearman Rho test, as well as a possible association was observed between exposure time and a lower risk of time-loss injury (R^2^ = 0.645) in lineal regression analysis
Watson et al. [25] Prospective USA 2 seasons	*n* = 19 NCAA D1 All male No age information	Time-loss injury: recorded by the team athletic trainer	Ratings of fatigue Ratings of mood Rating of soreness Rating of stress Rating of sleep quality Sleep duration (hours) sRPE	Nil	In the initial prediction models that were conducted separately, several factors were found to be significantly predictive of in-season injury. These factors included mood, fatigue, stress, soreness, and sleep duration (*p* < 0.001 for all), with odds ratios ranging from 0.41 to 0.57. However, in the subsequent multivariable models, only sleep duration and soreness remained significant, independent predictors, with odds ratios ranging from 0.52 to 0.69 and 0.65, respectively (*p* < 0.001 and *p* = 0.024, respectively). Mood, fatigue, and stress were no longer significant predictors, with odds ratios ranging from 1.1 to 1.2 and *p* values ranging from 0.43 to 0.69.
Benson et al. [23] Prospective Canada 1 season	*n* = 49 25 males 24 females High school teams in Calgary, AB Age 16.5 ± 0.6	Medical attention/time-loss injury: any physical complaint, including pain, ache, joint instability, stiffness, or any other complaint resulting from participating in basketball-related activities	sRPE	Jump count (no.) Jump height (cm) Weighted jump height (cm)	A low workload accumulation over 3 and 4 weeks coupled with a high 1-week workload could contribute to injury risk
Doeven et al. [22] Prospective Netherlands 1 season	*n* = 16 Dutch Basketball League All male Age 24.8 ± 2.0	Self-reported injury: Oslo Sports Trauma Research Center Questionnaire	sRPE Training sRPE Well-being Fatigue Sleep quality General muscle soreness Stress levels Mood Total quality of recovery (TQR)	Total load from training and matches Duration (min) Training load Training duration (min) Counter-movement jump height (cm)	No significant differences for severity scores and time loss were observed between short-term match congestion and regular competition
Ferioli et al. [17] Prospective Italy 2 seasons	*n* = 35 Italian Basketball League (D1-D3) All male Age 24 ± 6	Time-loss injury (non-contact injuries only): when a player was unable to fully take part in future basketball training or match due to physical complaints	sRPE Weekly RPE	Weekly load Week-to-week load change Duration (min) ACWR (1:2, 1:3, 1:4) (rolling average)	The study did not find any significant associations between the load markers and non-contact injuries (all *p* > 0.05). Additionally, the load markers exhibited no ability to predict injuries, as evidenced by the low Area Under the Curve (AUC) range of 0.468 to 0.537 and Youden index range of 0.019 to 0.132.
Garcia et al. [18] Prospective Spain 1 season	*n* = 8 Pardinyes competed in the “Leb plata” category All male Age 23.5 ± 2.56	A time-loss injury in basketball refers to a physical ailment sustained by a player during a match or training, caused by excessive transfer of energy that surpasses the body’s ability to maintain its structural and/or functional integrity. Such injuries result in the player being unable to fully participate in future basketball training or match play.	RPE sRPE	Total sessions (no. + min) Practice sessions (no. + min) Matches (no. + min)	A directly proportional but statistically non-significant relationship was observed in the connection between microtrauma injuries and RPE (F = 3.492; *p* = 0.112), but there is a directly proportional and statistically significant association between the team’s RPE and the one perceived by the coach (r = 0.775; *p* < 0.001)
Orringer & Pandya [26] Retrospective USA 3 seasons	*n* = 34 NBA All male Age 26.6 ± 4.89	Significant in-game injury leading to missing at least 10 consecutive games from Spotrac.com (accessed on 21 July 2021)	Nil	MPG	A higher number of minutes played per game in the three (4.9% increase, *p* = 0.04), five (5.8% increase, *p* = 0.004), and ten (4.0% increase, *p* = 0.02) games prior to the injury were significantly associated with a greater likelihood of injury occurrence.
Seibel et al. [28] Prospective USA 1 season	*n* = 16 NCAA D1 All female No age information	Injury data were extracted from medical injury reports generated as injuries occurred	Training Sleep Recovery Stress	Total weekly training load Daily average training load SD of weekly training load Weekly resistance training load	The study found that rapid eye movement (REM) sleep was the most significant contributor to injuries, with a 0.11 correlation coefficient for CORR, 2.7% for XGB, and 12.9% for RFC models. Additionally, low (<20%) and high (>30%) percentages of REM sleep increase the likelihood of injury. The partial dependence plots (PDPs) indicated that sleep disturbances increase when the respiratory rate falls outside the typical range of 12–18 repetitions per minute. Consequently, this increases the risk of injury.
Menon et al. [27] Retrospective USA 5 seasons	*n* = 196 NBA All male No age information	Season-ending injuries (SEIs) from Pro Sports Transactions: any injury that resulted in failure to return at least 5 games before the end of the team’s game schedule	Nil	MPG	A SEIs was significantly associated with minutes per game (odds ratio, 1.06, 95% confidence interval, 1.04–1.08, *p* < 0.001)

D: division; ACWR: acute/chronic workload ratio; RPE: rate of perceived exertion; sRPE: session of rate of perceived exertion; SD: standard deviation.

**Table 3 healthcare-12-01829-t003:** Quality of included studies as assessed on the Newcastle–Ottawa Scale (NOS).

Study	NOS Score				Level of Evidence
	Selection	Comparability	Outcome	Total Scores	
Anderson et al. (2003) [24]	3	1	2	6G	1
Gianoudis et al. (2008) [30]	2	0	2	4P	0
Caparrós et al. (2016) [19]	4	1	2	7G	1
Weiss et al. (2017) [29]	3	1	1	5P	1
Caparrós et al. (2018) [20]	3	1	3	7G	1
Piedra et al. (2020) [21]	4	1	2	7G	1
Watson et al. (2020) [25]	3	2	3	8G	2
Benson et al. (2021) [23]	3	2	3	8G	2
Doeven et al. (2021) [22]	3	1	1	5P	1
Ferioli et al. (2021) [17]	3	1	3	7G	1
Garcia et al. (2022) [18]	3	1	1	5P	1
Orringer and Pandya (2022) [26]	3	1	3	7G	1
Senbel et al. (2022) [28]	3	2	3	8G	2
Menon et al. (2024) [27]	3	1	2	6G	1
Median (range)	3 (2–4)	1 (0–2)	2 (1–3)	6 (4–8)	1 (0–2)

## Data Availability

Dataset available on request from the authors.

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
