# Peer review of "The Relationship between Training Load and Injury Risk in Basketball: A Systematic Review"

_healthcare, 2024, doi:10.3390/healthcare12181829_

Round 1

Reviewer 1 Report

Comments and Suggestions for Authors

I congratulated the authors for their study because the relationship between training load and injury in basketball is essential in sports injury prevention and performance enhancement. However, there is limited conclusive evidence of their associations. In the current study, the authors aimed to examine the evidence of the relationship between training load and injury in basketball, one of the most common sports worldwide. I want to state that I found the study academically successful and publishable. However, I have stated my opinions and suggestions regarding the study below to improve the quality of the study.

The weakest aspect of this study is the low number of studies included in the review. The number of publications should  be increased by increasing the inclusion criteria a little more. In addition, although the relationship between training load and injury was investigated in this study, the main logic and research content of the study focused on the increase in load due to the intensity of the competition calendar. Accordingly, either the title of the study should be changed or the content can be rearranged.

Finally, in order to be able to say that the injury is caused by overload, the injuries must be selected from non-contact injury types. It is recommended that authors state this in the limitations section or include studies that only cover non-contact injuries in the review.

Author Response

Reviewer 1

I congratulated the authors for their study because the relationship between training load and injury in basketball is essential in sports injury prevention and performance enhancement. However, there is limited conclusive evidence of their associations. In the current study, the authors aimed to examine the evidence of the relationship between training load and injury in basketball, one of the most common sports worldwide. I want to state that I found the study academically successful and publishable. However, I have stated my opinions and suggestions regarding the study below to improve the quality of the study.

Response: Thank you very much for your appreciation. I am deeply interested in basketball and load monitoring.

The weakest aspect of this study is the low number of studies included in the review. The number of publications should be increased by increasing the inclusion criteria a little more. In addition, although the relationship between training load and injury was investigated in this study, the main logic and research content of the study focused on the increase in load due to the intensity of the competition calendar. Accordingly, either the title of the study should be changed or the content can be rearranged.

Response: Thank you very much for your advice. There was limited number of articles examining the training load and injury risk in basketball. Therefore, the inclusion criteria have already increased to cover a wide range of training load and different kinds of musculoskeletal injury. I agree the main logic on increase in load, and therefore increase injury risk. The flow of main content was based on the magnitude and importance of the critical risk factors of training load.

Finally, in order to be able to say that the injury is caused by overload, the injuries must be selected from non-contact injury types. It is recommended that authors state this in the limitations section or include studies that only cover non-contact injuries in the review.

Response: Thank you very much for your advice. According to the included studies, some of the injuries were contacted type and some of the injuries were non-contact. The corresponding limitation has been added. “Fifth, most injuries were self-reported and non-contact types.”

Reviewer 2 Report

Comments and Suggestions for Authors

Avoid the use of acronyms in the abstract PRISMA and NOS is not needed.

Lines 29-31: you write in the past tense but what you state is still true in the present day, you should write it in the present simple tense.

Line 35: the acronym FIBA only appears twice in the manuscript; abbreviation should be avoided.

Line 46: Torres-Ronda and co-workers should be “Torres-Ronda et al.” and should be referenced after it: “Torres-Ronda et al. [6]”. The same in lines 178 and 280.

Line 58: IOC abbreviation is not needed as it only appears once in the whole text.

Lines 187 and 188. When you write “One study”, you must reference it there “One study [X]”.

Line 196: “Acute: Chronic Workload Ratio” correct this.

Line 206: “one studies” check this.

Lines 208-201: you must reference the studies you are talking about.

Table 2: What do you mean when you write “Nil”?

Although it is a good work, I miss that the conclusions are somewhat more concrete. I miss a range of MPG in which it can be said that this load decreases or increases the risk of injury, as you stated with the ACWR.

Author Response

Reviewer 2

Avoid the use of acronyms in the abstract PRISMA and NOS is not needed.

Response: Thank you very much. The acronyms in the abstract PRISMA and NOS are deleted.

Lines 29-31: you write in the past tense but what you state is still true in the present day, you should write it in the present simple tense.

Response: Thank you very much. The paragraph has been revised as follows. Lines 29-31: Load management, including prescription, monitoring, and adjustment of external and internal loads, is essential for enhancing performance and reducing injury risk for athletes [1]. In basketball, there is around 10 times higher injury incidence during competition than during training in professional basketball [2].

Line 35: the acronym FIBA only appears twice in the manuscript; abbreviation should be avoided.

Response: Thank you very much. The abbreviation of FIBA is deleted.

Line 46: Torres-Ronda and co-workers should be “Torres-Ronda et al.” and should be referenced after it: “Torres-Ronda et al. [6]”. The same in lines 178 and 280.

Response: Sorry for the mistake. The following changes have been made throughout the manuscript as follows:

“Torres-Ronda et al. [6]” in line 46,

“Ferioli et al. [17]” in line 178, and

“Gianoudis et al. [32]” in line 280.

Line 58: IOC abbreviation is not needed as it only appears once in the whole text.

Thank you very much. The abbreviation of IOC is deleted.

Lines 187 and 188. When you write “One study”, you must reference it there “One study [X]”.

Sorry for the mistake. The following changes have been made throughout the manuscript as follows:

“One study (6.25%) [31]” in line 187, and

“One study (6.25%) [32]” in line 188.

Line 196: “Acute: Chronic Workload Ratio” correct this.

Response: Thank you very much. ACWR in Line 196 has been deleted.

Line 206: “one studies” check this.

Response: Thankyou very much. Line 206 has been double checked.

Lines 208-201: you must reference the studies you are talking about.

Response: Thank you very much. The following changes have been made throughout the manuscript as follows. Lines 208-210: Studies also utilized RPE (n = 3) [17-18, 22], recovery and stress questionnaires/ratings (n = 3) [25, 28, 31], sleep (n =2) [25, 31], as well as ACWR (n = 2) [17, 24].

Table 2: What do you mean when you write “Nil”?

Response: Thank you very much for your comment. In table 2, there is nothing in the corresponding area.

Although it is a good work, I miss that the conclusions are somewhat more concrete. I miss a range of MPG in which it can be said that this load decreases or increases the risk of injury, as you stated with the ACWR.

Response: Thank you very much for your appreciation and comment. The results cannot be generalized because one of the studies compared the MPG to the season average and the other one mentioned the MPG in the condensed schedules.

Reviewer 3 Report

Comments and Suggestions for Authors

General comments

I would like to express my gratitude to the authors for the opportunity to review their systematic review. I find the manuscript to be generally well-written and aligned with the objectives and scope of the journal. However, there are several points that should be considered.

Specific comments

L2: The term "injury" is overly general. Could you specify whether it pertains to injury prevalence, injury risk, or another concept? Kindly ensure this specification is consistently applied throughout the manuscript, including the abstract, introduction, and discussion sections.

L95. Selection Criteria should be presented more clearly. The inclusion and exclusion criteria, as well as their definitions, should be explicitly identified.

L114. The Quality Assessment section requires more information. Who conducted the assessment? What was the process?

L116. A reference regarding the validation of the tool is needed.

L120. Please provide a reference explaining the rationale for establishing the different classification levels.

L130. Which researcher performed the data extraction? What methodology was used to minimize the risk of bias?

L155. Please provide references to the articles in the "Description of the Included Articles" section. This will help readers identify them quickly.

L193. The findings should be presented in greater depth.

L216. Please try to reduce the number of words in Table 2.

L220. "Assessment" is too general; please revise the section title.

L234. More information should be provided regarding the scores for each item in Table 3 and in the quality assessment section.

L237. The discussion section should begin with a brief paragraph summarizing the objectives and significance of the systematic review before delving into the discussion of each specific section.

L364. Including studies of varying competitive levels and genders could limit the generalizability of the results. Please mention this in both the discussion section and the limitations of your systematic review.

L384. What are the practical applications of this systematic review?

L386. Please review the grammar and punctuation in the conclusions section.

L387. The conclusions section should focus exclusively on the results of the SR.

L406. Some of the references do not include a DOI. Could you please provide it?

Author Response

Reviewer 3

L2: The term "injury" is overly general. Could you specify whether it pertains to injury prevalence, injury risk, or another concept? Kindly ensure this specification is consistently applied throughout the manuscript, including the abstract, introduction, and discussion sections.

Response: Thank you very much. The term “injury” refers to injury risk. It was defined by the International Olympic Committee in a consensus statement on load and risk of injury in 2016. The definition of injury was stated in the 2.2 Selection Criteria. i.e. “Injury was defined as any physical complaint resulting from competition or training, regardless of its impact on sports participation or performance. The term “injury” was replaced by “injury risk” throughout the manuscript, including the titles, abstract, introduction, and discussion.

L95. Selection Criteria should be presented more clearly. The inclusion and exclusion criteria, as well as their definitions, should be explicitly identified.

Response: Thank you very much for your comment. The selection criteria are mainly based on the current systematic reviews examining the training load and injury. The definitions of two key variables (i.e. training load and injury) are presented according to the consensus statement from the International Olympic Committee on load and risk of injury in 2016.

L114. The Quality Assessment section requires more information. Who conducted the assessment? What was the process? L130. Which researcher performed the data extraction? What methodology was used to minimize the risk of bias?

Response: Added “All three authors were involved in the assessment, the first author was responsible for the primary data extraction, the last author verified the data. If there was a disagreement, the second author would involve in the discussion.”

L116. A reference regarding the validation of the tool is needed.

L120. Please provide a reference explaining the rationale for establishing the different classification levels.

Response: Added reference “15.    Wells G,; Shea B,; O'Connell D,; Peterson J,; Welch V,; Losos M,; Tugwell, P. The Newcastle-Ottawa scale (NOS) for Assessing the Quality of Nonrandomised Studies in Meta-Analyses. University of Ottawa 2000.”

Response: Added reference “15.    Wells G,; Shea B,; O'Connell D,; Peterson J,; Welch V,; Losos M,; Tugwell, P. The Newcastle-Ottawa scale (NOS) for Assessing the Quality of Nonrandomised Studies in Meta-Analyses. University of Ottawa 2000.”

L155. Please provide references to the articles in the "Description of the Included Articles" section. This will help readers identify them quickly.

Response: Thank you very much for your comment. The corresponding references has been added to assist the readers as follows. All 14 of the included articles were cohort studies. The greatest number of articles were originated from Europe (n = 6, 43%) [17-18, 20-22, 25] and North America (n = 6, 43%) [16, 27-31], followed by Australia (n = 2, 14%) [24, 32]. Most studies (n = 8) (57%) [17-18, 21, 24-25, 28-30] had all male participants, three (21%) [22, 27, 31] had all females, and two (14%) [16, 32] had both. One (8%) [20] article did not specific the gender of participants. The mean age range was 16.0-27.6 years with three [28, 30-31] studies not reporting age. Most athletes from the studies (n = 9, 64.3%) [17-18, 20-22, 24-25, 29-30] played the top level of basketball league in their regions, with three studies (21.4%) [27, 28, 31] included college level athletes and two studies (14.3%) [16, 32] included high school level athletes accounting for the rest. Sample size ranged from 8 to 196, with seven studies (50%) [18, 22, 24-25, 27-28, 31] having less than 30 participants, six (42.9%) [16-17, 20-21, 29, 32] having 30-50 participants, one (7.1%) [30] having more than 100 participants. Eight studies (57.1%) [16, 18, 22, 24-25, 27, 31-32] monitored participants across 1 season (range from 16-34 weeks), two studies (14.3%) [17, 28] across 2 seasons, two studies (14.3%) [21, 29] across 3 seasons, two studies (14.3%) across 5 seasons [30] and 7 seasons [20] respectively. In terms of type of record, nine studies (64.3%) [16-18, 20, 22, 24-25, 28, 31] monitored training and competition, three studies (21.4%) [21, 29-30] monitored competitions, two studies (14.3%) monitored training [27] and monitored physical activity [32] respectively. Article characteristics are presented in Table 2.

L193. The findings should be presented in greater depth.

Response: Thank you for your advice. The detailed finding was presented in the discussion section.

L216. Please try to reduce the number of words in Table 2.

Response: Thank you for your advice. Some of the load measures in table 2 are deleted.

L220. "Assessment" is too general; please revise the section title.

Response: Thank you for your advice. The following changes have been made throughout the manuscript as follows. L220: Assessment of Article Quality, Level of Evidence, and Conflict of Interest

L234. More information should be provided regarding the scores for each item in Table 3 and in the quality assessment section.

Response: Thank you for your advice. There are only 3 parts of NOS scores for each study in table 3.

L237. The discussion section should begin with a brief paragraph summarizing the objectives and significance of the systematic review before delving into the discussion of each specific section.

Response: Thank you for your advice. The beginning of the discussion section has been added as follow. The primary purpose of this systematic review was to examine the evidence for a relationship between training load and injury risk in basketball. Our findings largely agree with the consensus statement from International Olympic Committee as well as the previous systematic review on load and injury risk [1, 11-13]. The results of this systematic review highlight the relationship between training load and injury risk in basketball as follows, although there were some limitations of generalizability of the results.

L364. Including studies of varying competitive levels and genders could limit the generalizability of the results. Please mention this in both the discussion section and the limitations of your systematic review.

Response: Thank you for your advice. The limitation of generalizability of the results has been added in the discussion and limitation sections.

L384. What are the practical applications of this systematic review?

Response: Thank you for your advice. The practical applications of this systematic review have been added. “For practitioners, they should regularly monitor the training load and sleep for athletes. Maintaining ACWR of 0.8-1.3 and 8-hours of sleep seems to be optimal based on the current evidence. In addition, they should manage the MPG of individual athlete to reduce the injury risk. Progressive increase MPG of athletes based on their season average seems to be optimal for injury prevention.”

L386. Please review the grammar and punctuation in the conclusions section.

Response: Sorry for the mistakes. The following changes have been made throughout the manuscript as follows. Line 387: To conclude, there was clear evidence in relationship between training load and injury in basketball, especially in sleep for the internal load monitoring and MPG for the external load monitoring.

L387. The conclusions section should focus exclusively on the results of the SR.

Response: Thank you for your advice. The conclusion section has been double checked. The statements are mainly based on the results of the systematic review.

L406. Some of the references do not include a DOI. Could you please provide it?

Response: Thank you for your comment. References 20, 21, 32, 34, has been double checked. Those references do not provide DOI.

Round 2

Reviewer 2 Report

Comments and Suggestions for Authors

All my comments have been ammended.

Author Response

Thank you very much for your support and wonderful review report.

Reviewer 3 Report

Comments and Suggestions for Authors

The format in which the authors present the changes is unclear. If possible, please enable the automatic change-tracking feature or highlight the revised text in yellow.

L116. A reference regarding the validation of the tool is needed.

L120. Please provide a reference explaining the rationale for establishing the different classification levels.

Response: Added reference “15.    Wells G,; Shea B,; O'Connell D,; Peterson J,; Welch V,; Losos M,; Tugwell, P. The Newcastle-Ottawa scale (NOS) for Assessing the Quality of Nonrandomised Studies in Meta-Analyses. University of Ottawa 2000.”

Response: Added reference “15.    Wells G,; Shea B,; O'Connell D,; Peterson J,; Welch V,; Losos M,; Tugwell, P. The Newcastle-Ottawa scale (NOS) for Assessing the Quality of Nonrandomised Studies in Meta-Analyses. University of Ottawa 2000.”

Comment: Please, cite the reference in section 2.3. (Quality assessment).

L193. The findings should be presented in greater depth.

Response: Thank you for your advice. The detailed finding was presented in the discussion section.

Comment:  The results should be presented in greater depth in the results section, rather than in the discussion section.

L384. What are the practical applications of this systematic review?

Response: Thank you for your advice. The practical applications of this systematic review have been added. “For practitioners, they should regularly monitor the training load and sleep for athletes. Maintaining ACWR of 0.8-1.3 and 8-hours of sleep seems to be optimal based on the current evidence. In addition, they should manage the MPG of individual athlete to reduce the injury risk. Progressive increase MPG of athletes based on their season average seems to be optimal for injury prevention.”

Comment:  The practical applications should appear in the discussion section or in a subsection titled "Practical Applications," rather than in the "Conclusions" section. Additionally, they should be based solely on the results of the present discussion, not on the authors' recommendations.

Author Response

The format in which the authors present the changes is unclear. If possible, please enable the automatic change-tracking feature or highlight the revised text in yellow.

Response: The revised texts are now highlighted in yellow and shown in change-tracking. Sorry for the inconvenience caused.

L116. A reference regarding the validation of the tool is needed.

L120. Please provide a reference explaining the rationale for establishing the different classification levels.

Comment: Please, cite the reference in section 2.3. (Quality assessment).

Response: The citation is now highlighted in red. Sorry for the inconvenience caused.

Added reference “15.    Wells G,; Shea B,; O'Connell D,; Peterson J,; Welch V,; Losos M,; Tugwell, P. The Newcastle-Ottawa scale (NOS) for Assessing the Quality of Nonrandomised Studies in Meta-Analyses. University of Ottawa 2000.”

L193. The findings should be presented in greater depth.

Comment:  The results should be presented in greater depth in the results section, rather than in the discussion section.

Response: Thank you for your suggestion. After careful discussion between the authors, we decided to add 60 words to present the findings in greater depth.

Added “Training load in basketball was measured using internal loads, external loads, or both, with the majority of studies utilizing absolute measures and some incorporating the ACWR as a relative measure. Overall, most studies found a relationship between training load and injury risk, with some indicating a direct relationship and others showing both direct and inverse relationships, while a few studies reported no significant relationship.”

L384. What are the practical applications of this systematic review?

Comment:  The practical applications should appear in the discussion section or in a subsection titled "Practical Applications," rather than in the "Conclusions" section. Additionally, they should be based solely on the results of the present discussion, not on the authors' recommendations.

Response: Thank you so much for your advice to improve the manuscript. We agreed and revised as below.

Added in section 4.8 practical applications and future direction “Practically, evidence shows that increasing sleep duration and maintaining optimal sleep patterns, along with managing MPG, are key for injury prevention in basketball. Practitioners should regularly monitor training load and sleep, aim for an ACWR of 0.8-1.3, ensure athletes get 8 hours of sleep, and progressively increase MPG based on the season average to reduce injury risk.”